# The Effects of Alcohol Intoxication and Withdrawal on Hypothalamic Neurohormones and Extrahypothalamic Neurotransmitters

**DOI:** 10.3390/biomedicines11051288

**Published:** 2023-04-27

**Authors:** Balázs Simon, András Buzás, Péter Bokor, Krisztina Csabafi, Katalin Eszter Ibos, Éva Bodnár, László Török, Imre Földesi, Andrea Siska, Zsolt Bagosi

**Affiliations:** 1Department of Pathophysiology, Albert Szent-Györgyi Medical School, University of Szeged, Semmelweis Str. 1, 6720 Szeged, Hungary; 2Department of Traumatology, Albert Szent-Györgyi Medical School, University of Szeged, 6720 Szeged, Hungary; 3Institute of Laboratory Medicine, Albert Szent-Györgyi Medical School, University of Szeged, 6720 Szeged, Hungary

**Keywords:** alcohol intoxication, alcohol withdrawal, CRF, AVP, dopamine, GABA, glutamate

## Abstract

The aim of the present study was to determine the effects of alcohol intoxication and withdrawal on hypothalamic neurohormones such as corticotropin-releasing factor (CRF) and arginine vasopressin (AVP), and extrahypothalamic neurotransmitters such as striatal dopamine (DA), amygdalar gamma aminobutyric acid (GABA), and hippocampal glutamate (GLU). In addition, the participation of the two CRF receptors, CRF1 and CRF2, was investigated. For this purpose, male Wistar rats were exposed to repeated intraperitoneal (ip) administration of alcohol every 12 h, for 4 days and then for 1 day of alcohol abstinence. On the fifth or sixth day, intracerebroventricular (icv) administration of selective CRF1 antagonist antalarmin or selective CRF2 antagonist astressin_2_B was performed. After 30 min, the expression and concentration of hypothalamic CRF and AVP, the concentration of plasma adrenocorticotropic hormone (ACTH) and corticosterone (CORT), and the release of striatal DA, amygdalar GABA, and hippocampal GLU were measured. Our results indicate that the neuroendocrine changes induced by alcohol intoxication and withdrawal are mediated by CRF1, not CRF2, except for the changes in hypothalamic AVP, which are not mediated by CRF receptors.

## 1. Introduction

Alcohol addiction has three stages including binge/intoxication, withdrawal/negative affect, and preoccupation/anticipation (craving). Each stage is characterized by specific changes in the hypothalamic neurohormones such as corticotropin-releasing factor (CRF) and arginine vasopressin (AVP), as well as extrahypothalamic neurotransmitters such as striatal dopamine (DA), amygdalar gamma aminobutyric acid (GABA), and hippocampal glutamate (GLU) [1,2].

The stage of binge/intoxication is associated with the activation of the hypothalamic–pituitary–adrenal (HPA) axis that is initiated by hypothalamic CRF and/or AVP [3]. The HPA axis consists of the paraventricular nucleus of the hypothalamus (PVN), the anterior pituitary, and the adrenal cortex, and can be stimulated by various stressors [4,5]. Alcohol can also stimulate the release of CRF and/or AVP from the PVN, which, in turn, evokes the release of adrenocorticotrop hormone (ACTH) from the anterior pituitary. Subsequently, pituitary ACTH stimulates the release of glucocorticoids in the adrenal cortex, which are represented mainly by cortisol in humans and corticosterone (CORT) in rodents. The elevation of the plasma ACTH and glucocorticoid levels not only reflects the activation of the HPA axis, but also exerts a negative feedback effect on the release of hypothalamic CRF and/or AVP, thereby inhibiting the HPA axis [4,5].

The stage of withdrawal/negative affect is associated with the activation of the extended amygdala circuit, which is mediated by extrahypothalamic CRF and norepinephrine [3]. The extended amygdala circuit consists of the central nucleus of the amygdala (CeA), the bed nucleus of stria terminalis (BNST), and the shell of the nucleus accumbens (shNAcc), and represents an interface between the reward and stress systems [4,5]. During alcohol intoxication, the reward system is activated [3]. Alcohol stimulates the DA release in the striatum, and the GABA release in the amygdala, inducing rewarding, anxiolytic, and antidepressant effects [3]. Alcohol intoxication may also result in amnesia, which can be related to the reduction in the hippocampal GLU release [3]. During alcohol withdrawal, as an anti-reward mechanism, the stress system is activated, resulting in alcohol withdrawal syndrome, which includes symptoms such as anhedonia, anxiety, and aggression [3]. These symptoms can be explained by the reduction in the striatal DA and amygdalar GABA release, and the stimulation of GLU release in the hippocampus [3].

Alcohol withdrawal syndrome consists of somatic (physical) signs and affective (emotional) symptoms that emerge immediately after alcohol cessation [1,2]. The physical signs usually cease within 24 h following alcohol intoxication (acute alcohol withdrawal), whereas the emotional symptoms may persist for days to years (protracted alcohol withdrawal), producing craving that makes one vulnerable to relapse, especially in periods of stress [1,2]. This last stage of preoccupation/anticipation (craving) is associated with the activation of the hippocampus, orbitofrontal cortex, prefrontal cortex, insula and basolateral amygdala (BLA), and is believed to be mediated by both hypothalamic and extrahypothalamic CRF [3].

CRF acts via two distinct CRF receptors, CRF1 and CRF2, with putatively antagonistic actions in the central nervous system (CNS) [6]. CRF1 is distributed predominantly in the cerebral cortex, anterior pituitary, and cerebellum, but it is also expressed in the striatum, amygdala, and hippocampus [7], and seems to promote activation of the HPA axis, anxiety, and depression [8,9,10]. In contrast, CRF2 is limited centrally to the subcortical regions including the striatum, amygdala, and hippocampus [7] and appears to mediate anxiolytic and antidepressant effects [8,9,10].

The aim of our study was to determine the effects of alcohol intoxication and withdrawal on hypothalamic neurohormones and extrahypothalamic neurotransmitters in rats. Since CRF was involved in all three stages of alcohol addiction, the participation of the two CRF receptors was investigated. For this purpose, male Wistar rats were exposed to repeated intraperitoneal (ip) administration of alcohol every 12 h, for 4 days and then for 1 day of alcohol withdrawal. On the fifth day (immediately after the last ip administration of alcohol) or the sixth day (24 h after the last ip administration of alcohol), intracerebroventricular (icv) administration of the selective CRF1 antagonist antalarmin or selective CRF2 antagonist astressin_2_B was performed. After 30 min, the mice were decapitated without anesthesia, trunk blood was collected, and the brains were removed. From the brain, the expression and concentration of hypothalamic CRF and AVP as well as the release of striatal DA, amygdalar GABA, and hippocampal GLU were determined. From the trunk blood, the concentration of plasma ACTH and CORT were measured.

## 2. Materials and Methods

### 2.1. Animals

Male Wistar rats weighing 150–250 g were used (*N* = 72). The rats were treated in accordance with the ARRIVE guidelines and the experiments were carried out in concordance with the EU Directive 2010/63/EU for animal experiments. They were housed together and kept in their home cages at a constant temperature on a standard illumination schedule with 12-h light and 12-h dark periods (lights on from 6:00). Commercial food and tap water were available ad libitum. They were also handled daily in order to minimize the effects of non-specific stress.

### 2.2. Substances

The saline and alcohol solutions administered ip were provided by B. Braun Inc., Melsungen, Germany and Reanal Ltd., Budapest, Hungary, respectively. The selective CRF1 antagonist antalarmin and the selective CRF2 antagonist astressin_2_B administered via icv administration were ordered from Sigma-Aldrich Inc., St. Louis, MO, USA. The acetic acid used for the in vitro homogenization was provided by Reanal Ltd., Budapest, Hungary. Ethanol was determined from the plasma by commercially available enzymatic kit (Roche Diagnostics, Mannheim, Germany) on cobas c502 analyzer (Roche Diagnostics, Mannheim, Germany). The sensitivity of the assay was 10.1 mg/dL (0.01 g/dL). The GeneJET RNA Purification Kit, the Maxima First Strand cDNA Synthesis Kit, and the NanoDrop One device required for the determination of hypothalamic CRF and AVP expression was provided by Thermo Scientific Inc., Waltham, MA, USA. The sandwich Enzyme-linked Immunosorbent Assay (ELISA) Kits required for the determination of the hypothalamic CRF and AVP, and plasma ACTH concentrations were purchased from Phoenix Pharmaceuticals Ltd., Mannheim, Germany. The methylene chloride, sulfuric acid, and ethyl-alcohol solutions used for the determination of the plasma CORT concentration were provided by Reanal Ltd., Budapest, Hungary. The Krebs solution was provided by Reanal Ltd., Hungary. The tritium-labelled neurotransmitters including the [^3^H]DA, [^3^H]GABA, and [^3^H]GLU, and the Ultima Gold scintillation fluid used for the in vitro superfusion studies and liquid scintillation were ordered from Perkin-Elmer Inc., Waltham, MA, USA.

### 2.3. Surgery

The rats were implanted with a stainless steel Luer cannula (10 mm long) that was aimed at the right lateral cerebral ventricle. The surgical intervention was performed under anesthesia with 35 mg/kg pentobarbital sodium (Euthanasol, CEVA-Phylaxia, Budapest, Hungary). The stereotaxic coordinates for the right lateral cerebral ventricle were 0.2 mm posterior and 1.7 mm lateral to the bregma, and 3.7 mm deep from the dural surface, according to a stereotaxic atlas of the rat brain [11]. Cannulas were secured to the skull with dental cement and acrylate. The rats were allowed to recover for 7 days after surgery.

### 2.4. Treatment

Male Wistar rats were exposed to repeated ip administration of alcohol every 12 h, for 4 days and then for 1 day of alcohol withdrawal. The protocol of alcohol administration was based on a previous study in which 20% alcohol was administered at dose of 3 g/kg [12] (Table 1).

This amount of alcohol produced a blood alcohol concentration (BAC) of 197.5 ± 19 mg/dL, measured at 30 min after the ip administration [12]. On the fifth (immediately after the last ip administration of alcohol) or sixth day (24 h after the last ip administration of alcohol), the rats were administered icv with 0.1 µg/2 µL of antalarmin or 1 µg/2 µL of astressin_2_B. The doses of CRF1 and CRF2 antagonists were based on our previous studies, which indicated that these doses efficiently block the activation of the HPA axis and the striatal DA release observed during nicotine withdrawal [13,14]. After 30 min, the mice were decapitated without anesthesia, the trunk blood was collected, and the brains were removed. From the brain, the expression and concentration of hypothalamic CRF and AVP as well as the release of striatal DA, amygdalar GABA, and hippocampal GLU were determined. From the trunk blood, the concentration of plasma ACTH and CORT were measured.

### 2.5. Quantitative Reverse Transcription Polymerase Chain Reaction (RT-qPCR)

For the determination of the hypothalamic CRF and AVP expression, quantitative reverse transcription polymerase chain reaction (RT-qPCR) was performed. First, the rats were decapitated, their brains removed, and then dissected in a Petri dish filled with ice-cold Krebs solution. The hypothalamus was isolated from each rat according to a stereotaxic atlas of the rat brain [11] after the following coordinates: rostro-caudal (RC) +2.6 to −2.6 mm, medio-lateral (ML) +1.5 to −1.5 mm, dorso-ventral (DV) +7 to +10 mm (Figure 1).

The tissue samples were stored in 1 mL of TRIzol (Thermo Fisher Scientific, Waltham, MA, USA) in Eppendorf tubes and kept in a freezer at −80 °C and then underwent ultrasonic homogenization (Branson Sonifier 250, Emerson, St. Louis, MO, USA), then 200 µL of chloroform was added to each sample. Following 10 min of incubation at room temperature, the samples were centrifuged for 15 min at 13,000× *g* (Heraeus Fresco 17, Thermo Fisher Scientific, Waltham, MA, USA). Approximately 500 µL of the supernatant was collected from each tube and transferred to new Eppendorf tubes containing 600 µL of 96% alcohol that were stored overnight at −20 °C. On the following day, the GeneJET RNA Purification Kit (Thermo Fisher Scientific, USA) was used according to the manufacturer’s instructions. The concentration of the purified samples was calculated based on the average of three measurements with a spectrophotometer (NanoDrop One, Thermo Scientific Inc., Waltham, MA, USA). The RNA samples were deemed uncontaminated if the 260/280 nm ratio was between 1.8 and 2.2. A volume containing 300 ng of RNA was obtained from each sample for cDNA synthesis. The first strand cDNA was synthesized using the Maxima First Strand cDNA Synthesis Kit (Thermo Scientific Inc., Waltham, MA, USA) according to the manufacturer’s instructions.

The qPCR reaction mix was prepared using the Luminaris Color HiGreen Low ROX qPCR Master Mix (Thermo Fisher Scientific, Waltham, MA, USA) according to the manufacturer’s instructions. A total volume of 10 µL of reaction mix was prepared containing 5 µL of Master Mix, 0.3 µL of forward primer, 0.3 µL of reverse primer, 1.67 µL of cDNA, and 2.73 µL of nuclease-free water. The custom primers corresponding to the CRF, AVP, and GPADH genes are shown in Table 2.

The mix was placed in a thermal cycler (C1000 Touch Thermal Cycler, BioRad, Budapest, Hungary) that was programmed according to the cycling protocol in Table 3. The expression of each gene relative to *Gapdh* was determined using the ΔΔCT method.

### 2.6. Sandwich Enzyme-Linked Immunosorbent Assay (ELISA)

For the determination of the hypothalamic CRF and AVP and plasma ACTH concentrations an in vitro homogenization was performed [15,16], which was followed by the sandwich ELISA. First, the rats were decapitated, their brains removed, and then dissected in a Petri dish filled with ice-cold Krebs solution and their trunk blood collected. The hypothalamus was isolated from each rat according to the stereotaxic atlas of the rat brain, as previously described [11]. The samples were dissolved in 500 µL acetic acid at a 2 M concentration in Eppendorf tubes and immersed in boiling water for 5 min. Next, the samples were homogenized with an ultrasonic homogenizer (Branson Sonifier 250, Emerson, St. Louis, MO, USA) on ice for 30 s. The homogenates were centrifuged twice at 10,000 rpm at 4 °C for 20 min, after which the supernatants were separated and underwent lyophilization for further determinations. The hypothalamic CRF and AVP and plasma ACTH concentrations were determined according to the manufacturer’s instructions (Phoenix Pharmaceuticals Ltd., Mannheim, Germany) and expressed as ng/mL.

### 2.7. Chemofluorescent Assay

For the determination of the plasma CORT concentration, a chemofluorescent assay was performed as described originally by Purves and Sirett, and modified later by Zenker and Bernstein [17,18]. All substances used, including methylene chloride, sulfuric acid, and ethyl-alcohol solutions, used for the determination of the plasma CORT concentration were provided by Reanal Ltd., Budapest, Hungary and the plasma CORT concentration was expressed as µg/100 mL.

### 2.8. In Vitro Superfusion Assay

The striatal DA, amygdalar GABA, and hippocampal GLU release were measured by means of an in vitro superfusion system, which was described originally by Gaddum, and later improved by Harsing and Vizi [19,20]. The rats were decapitated, their brains removed, and then dissected in a Petri dish filled with ice-cold Krebs solution. The striatum, amygdala, and hippocampus were isolated from each rat according to a stereotaxic atlas of the rat brain [11] after the following coordinates: RC +4.0 to −1.0 mm, ML +1.0 to +5.0 mm, DV +3.0 to +8.0 mm for the striatum (Figure 2); RC 0.0 to −2.0 mm, ML +3.0 to +6.0 mm, DV +7.0 to +10.0 mm for the amygdala (Figure 3); and RC −4.0 to −6.0 mm, ML +2.0 to +5.0 mm, DV +3.0 to +8.0 mm for the hippocampus (Figure 4). The brain tissue was dissected, incubated for 30 min in 8 mL of Krebs solution with 15 mM of [^3^H]DA, [^3^H]GABA, or [^3^H]GLU incubated, then superfused for 30 min and electrically stimulated for 2 min with the means of a superfusion system provided by Experimetria Ltd., Budapest, Hungary. The superfusates were collected in Eppendorf tubes by a multichannel fraction collector (Gilson FC 203B). The total collecting time was 32 min (4 × 16 samples, 2 min each) and the peak of the fractional release was observed at 14 min. In the corresponding figures, only these peaks were represented. Finally, the brain tissue was removed from the superfusion system and solubilized in Krebs solution using an ultrasonic homogenizer called a Branson Sonifier 250. After the addition of scintillation fluid to the samples and the remaining brain tissue, the radioactivity was measured with a liquid scintillation spectrometer (Tri-carb 2100TR, Packard Inc., Ramsey, MN, USA) and expressed in count per minute (CPM). The fractional release was calculated as the ratio between the radioactivity of the samples and that of the remaining brain tissue.

## 3. Statistical Analysis

Data were presented as the means ± SEM. Statistical analysis of the results was performed by ANOVA, if the test prerequisites allowed, using SPSS Software v.29.0 (IBM Inc., Amonk, NY, USA). A two-way 2 (alcohol or saline) × 3 (Antalarmin or Astressin_2_B or saline) ANOVA was performed with the estimated marginal means calculated followed by the Bonferroni post hoc test. A probability level of less than 0.05 was accepted as indicating a statistically significant difference.

## 4. Results

Hypothalamic CRF mRNA expression was increased by alcohol intoxication and withdrawal, and these stimulatory effects were reduced by antalarmin, but not astressin_2_B (Figure 5). On the fifth day, a significant main effect in the alcohol-treated groups [F(1,17) = 3.568, *p* < 0.001], a significant main effect in the antagonist-treated group [F(2,11) = 1.875, *p* < 0.001], and a significant interaction between the two factors [F(2,17) = 2.02, *p* < 0.001] were observed. On the sixth day, a significant main effect in the alcohol-treated group [F(1,17) = 12.476, *p* < 0.001], a significant main effect in the antagonist-treated group [F(2,11) = 8.535, *p* < 0.001], and a significant interaction between the two factors [F(2,17) = 17.435, *p* < 0.001] were detected.

In addition, the hypothalamic CRF concentration was increased by alcohol intoxication and withdrawal, and these stimulatory effects were reversed by the selective CRF1, but not CRF2 antagonist (Figure 6). On the 5th day, a significant main effect in the alcohol-treated group [F(1,134) = 39.173, *p* < 0.001], but no significant main effect in the antagonist-treated group, yet a significant interaction between the two factors [F(1,17) = 3.356, *p* < 0.001] were assessed. On the 6th day, a significant main effect in the alcohol-treated group [F(1,35) = 10.021, *p* = 0.004], but no significant main effect in the antagonist-treated group and no significant interaction between the two factors were noticed.

Hypothalamic AVP mRNA expression was decreased by alcohol intoxication and withdrawal, but these inhibitory effects were not significantly influenced either by antalarmin or astressin_2_B (Figure 7). On the fifth day, a significant main effect in the alcohol-treated group [F(1,17) = 3.451, *p* < 0.001], no significant main effect in the antagonist-treated group, but a significant interaction between the two factors [F(2,11) = 6.299, *p* < 0.001] were observed. On the sixth day, a significant main effect in the alcohol-treated groups [F(1,17) = 6.425, *p* < 0.001], no significant main effect in the antagonist-treated group, but a significant interaction between the two factors [F(2,11) = 10.245, *p* < 0.001] were detected.

In contrast, hypothalamic AVP concentration was increased by alcohol intoxication and withdrawal, but these stimulatory effects were not affected by none of the selective CRF antagonists (Figure 8.). On the 5th day, a significant main effect in the alcohol-treated group [F(1,35) = 85.4, *p* < 0.001], but no significant main effect in the antagonist-treated group, and no significant interaction between the two factors were assessed. On the 6th day, a significant main effect in the alcohol-treated group [F(1,35) = 80.680, *p* < 0.001], but no significant main effect in the antagonist-treated group, and no significant interaction between the two factors were noticed.

The plasma ACTH level was elevated by alcohol intoxication and withdrawal, and these elevations were ameliorated by antalarmin, but not astressin_2_B (Figure 9). On the fifth day, a significant main effect in the alcohol-treated group [F(1,35) = 64.352, *p* < 0.001], a significant main effect in the antagonist-treated group [F(2,35) = 12.523, *p* < 0.001], and a significant interaction between the two factors [F(2,35) = 9.311, *p* < 0.001] were proved. On the 6th day, a significant main effect in the alcohol-treated group [F(1,35) = 49.394, *p* < 0.001], but no significant main effect in the antagonist-treated group, and no significant interaction between the two factors were shown.

In parallel, the plasma CORT level was augmented by alcohol intoxication and withdrawal, and these augmentations were attenuated by the selective CRF1, but not CRF2 antagonist (Figure 10). On the fifth day, a significant main effect in the alcohol-treated group [F(1,35) = 30.996, *p* < 0.001], a significant main effect in the antagonist-treated group [F(2,35) = 3.892, *p* < 0.001], and a significant interaction between the two factors [F(2,35) = 4.778, *p* < 0.001] were shown. On the 6th day, a significant main effect in the alcohol-treated group [F(1,35) = 62.363, *p* < 0.001], a significant main effect in the antagonist-treated group [F(2,35) = 9.192, *p* < 0.001], and a significant interaction between the two factors [F(2,35) = 9.765, *p* < 0.001] were proved.

Alcohol intoxication increased, whereas alcohol withdrawal decreased the striatal DA release, and both effects were reduced by antalarmin, but not astressin_2_B (Figure 11). On the fifth day, a significant main effect in the alcohol-treated group [F(1,17) = 271.724, *p* < 0.001], a significant main effect in the antagonist-treated group [F(2,11) = 98.352, *p* < 0.001], and a significant interaction between the two factors [F(2,11) = 100.708, *p* < 0.001] were observed. On the sixth day, a significant main effect in the alcohol-treated group [F(1,17) = 319.151, *p* < 0.001], a significant main effect in the antagonist-treated group [F(2,11) = 52.758, *p* < 0.001], and a significant interaction between the two factors [F(2,11) = 103.865, *p* < 0.001] were detected.

Similarly, alcohol intoxication increased, whereas alcohol withdrawal decreased the amygdalar GABA release and both effects were reversed by the selective CRF1, but not CRF2 antagonist (Figure 12). On the fifth day, a significant main effect in the alcohol-treated group [F(1,17) = 226.989, *p* < 0.001], a significant main effect in the antagonist-treated group [F(2,11) = 68.927, *p* < 0.001], and a significant interaction between the two factors [F(2,11) = 57.735, *p* < 0.001] were assessed. On the sixth day, a significant main effect in the alcohol-treated group [F(1,17) = 123.070, *p* < 0.001], a significant main effect in the antagonist-treated group [F(2,11) = 36.792, *p* < 0.001], and a significant interaction between the two factors [F(2,11) = 60.402, *p* < 0.001] were noticed.

Consequently, the hippocampal GLU release was decreased and increased in alcohol intoxication and alcohol withdrawal, respectively, and both effects were antagonized by antalarmin, but not astressin_2_B (Figure 13). On the fifth day, a significant main effect in the alcohol-treated group [F(1,17) = 663.538, *p* < 0.001], a significant main effect in the antagonist-treated group [F(2,11) = 172.546, *p* < 0.001], and a significant interaction between the two factors [F(2,11) = 257.706, *p* < 0.001] were shown. On the sixth day, a significant main effect in the alcohol-treated group [F(1,17) = 305.855, *p* < 0.001], a significant main effect in the antagonist-treated group [F(2,11) = 97.827, *p* < 0.001], and a significant interaction between the two factors [F(2,11) = 82.538, *p* < 0.001] were proved.

## 5. Discussion

Our results indicate that during alcohol intoxication, the HPA axis is activated by hypothalamic CRF and is reflected by the elevation of plasma CORT and ACTH levels. Aside from the stress axis, the reward system is also activated, mainly resulting in increased striatal DA, but also increased amygdalar GABA and decreased hippocampal GLU release. During alcohol withdrawal, the HPA axis remains activated, but this time, the activation is accompanied by the decrease in the striatal DA and amygdalar GABA release, and increase in hippocampal GLU release, which is probably mediated by extrahypothalamic CRF.

Regarding the hypothalamic neurohormones, previous studies have already suggested the role of CRF in the HPA-axis activation induced by alcohol based on several lines of evidence [21]. First, the administration of a CRF antiserum or a CRF antagonist inhibited the stimulatory effect of alcohol on ACTH secretion in rats [22,23,24]. Second, bilateral destruction of the paraventricular CRF-secreting neurons ameliorated, although did not abolish, the alcohol-stimulated ACTH secretion [25,26]. Third, the administration of alcohol increased the expression of CRF heteronuclear RNA and mRNA levels as well as the expression of c-Fos mRNA and the Fos protein in the PVN [25,27,28]. Furthermore, the present study emphasizes the role of hypothalamic CRF in the activation of the HPA axis during alcohol intoxication and withdrawal. This finding is supported by the observation that both the expression and concentration of the hypothalamic CRF increased in parallel with the levels of plasma ACTH and CORT, immediately and 24 h after the last alcohol administration. Previous studies have also suggested the role of AVP in the alcohol-induced activation of the HPA-axis [21]. First, the administration of an AVP antiserum or an AVP antagonist inhibited the stimulatory effect of alcohol on ACTH secretion in rats [24,29]. Second, the removal of endogenous AVP in rats previously exposed to bilateral destruction of the paraventricular neurons partially diminished the alcohol-stimulated ACTH secretion [25]. Third, the administration of alcohol increased the expression of AVP heteronuclear RNA and mRNA levels [27]. However, the present study questions the role of the hypothalamic AVP in activating the HPA axis during alcohol intoxication and withdrawal. This is based on the speculation that the decreased hypothalamic AVP expression and the increased hypothalamic AVP concentration immediately and 24 h after the last alcohol administration represents a decreased release rather than an increased synthesis of hypothalamic AVP, a process that might have resulted from the negative feedback of plasma glucocorticoids and ACTH, or could be related to another function of AVP such as water retention.

Regarding the extrahypothalamic neurotransmitters, previous studies have suggested that alcohol binge/intoxication and withdrawal/negative effects are also associated with certain changes in the striatal DA, amygdalar GABA, and hippocampal GLU [1,2]. Acute alcohol administration stimulates the striatal DA release, inducing the sensation of reward, whereas chronic alcohol administration leads to a decrease in the DA release in the striatum, which is manifested in a reward deficit during alcohol withdrawal [30,31]. This reward deficit can be explained by an increase in the reward threshold caused by the downregulation of pre-synaptic DA receptors, and a decrease in extracellular DA release caused by the depletion of striatal DA stores that are demasked during alcohol withdrawal [32]. The amygdalar GABA is presumed to play a role in the positive, anxiolytic effects of alcohol [33,34,35,36]. Acute alcohol consumption facilitates GABA-ergic neurotransmission in CeA via both pre- and post-synaptic mechanisms, whereas chronic alcohol consumption increases the baseline GABA-ergic neurotransmission, but not the stimulated GABA release [37]. The hippocampal GLU is believed to play a role in the negative, anxiogenic effects of alcohol and the development of aggression observed especially during alcohol withdrawal [12,38,39,40]. In general, acute alcohol consumption decreases glutamatergic neurotransmission by the downregulation of N-methyl-d-aspartate (NMDA) and α-amino-3-hydroxy-5-methyl-4-isoxazolepropionic acid (AMPA) receptors, whereas chronic alcohol consumption increases glutamatergic neurotransmission by the upregulation of NMDA receptors and the stimulation of GLU release, which might be further enhanced by repeated periods of alcohol withdrawal [37].

As for the role of CRF receptors in the neurohormonal changes induced by alcohol, previous in vivo experiments have already reported that alcohol induced upregulation of paraventricular CRF1 expression [41], and that the alcohol-induced ACTH secretion could be blocked by the non-selective CRF receptor antagonist astressin and selective CRF1 antagonist NBI 30775 [42]. In contrast, alcohol administration was unable to produce the upregulation of CRF2 expression in the PVN [41], and the alcohol-induced ACTH secretion could not be blocked by the selective CRF2 antagonist astressin_2_B [42]. This is in agreement with our previous in vitro experiments that demonstrated that striatal DA and amygdalar GABA release could be stimulated or inhibited by non-selective CRF1 agonists or antagonists, but are not affected by selective CRF2 agonists or antagonists [43,44,45]. In accordance with the previous findings, our present study underlines the role of CRF1 in the changes in hypothalamic neurohormones and extrahypothalamic neurotransmitters observed during alcohol intoxication and withdrawal. Nevertheless, based on the present findings, the role of CRF receptors in the alcohol-induced alteration of the hypothalamic AVP can be excluded.

## 6. Conclusions

In conclusion, our results indicate that the neuroendocrine changes induced by alcohol intoxication and withdrawal are mediated by CRF1, not CRF2, except for the changes in hypothalamic AVP, which are not mediated by CRF receptors. Therefore, our study demonstrates that 4-day ip administration of alcohol followed by 1-day abstinence in rats is a valid model for alcohol intoxication and withdrawal, characterized by specific changes in hypothalamic neurohormones and extrahypothalamic neurotransmitters, which can be used for therapeutical purposes.

## Figures and Tables

**Figure 1 biomedicines-11-01288-f001:**
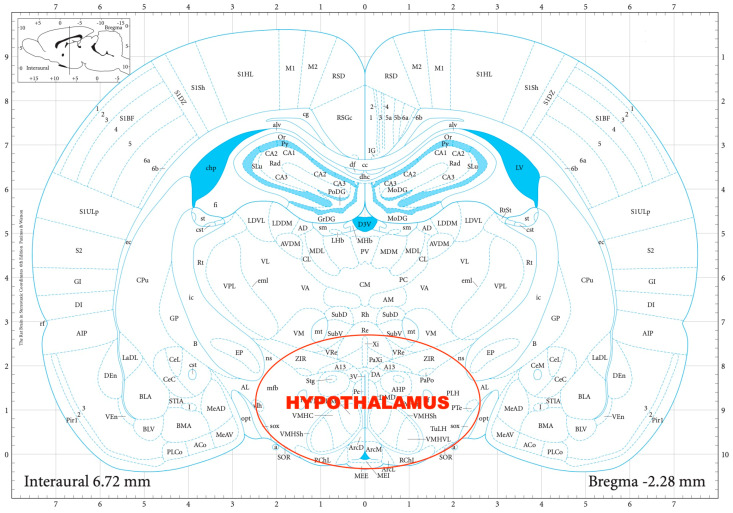
The dissection of the hypothalamus according to a stereotaxic atlas of the rat brain. Reprinted/adapted with permission from Ref. [11]. The coordinates were: rostro-caudal (RC) +2.6–−2.6 mm, medio-lateral (ML) +1.5–−1.5 mm, dorso-ventral (DV) +7–+10 mm for the hypothalamus.

**Figure 2 biomedicines-11-01288-f002:**
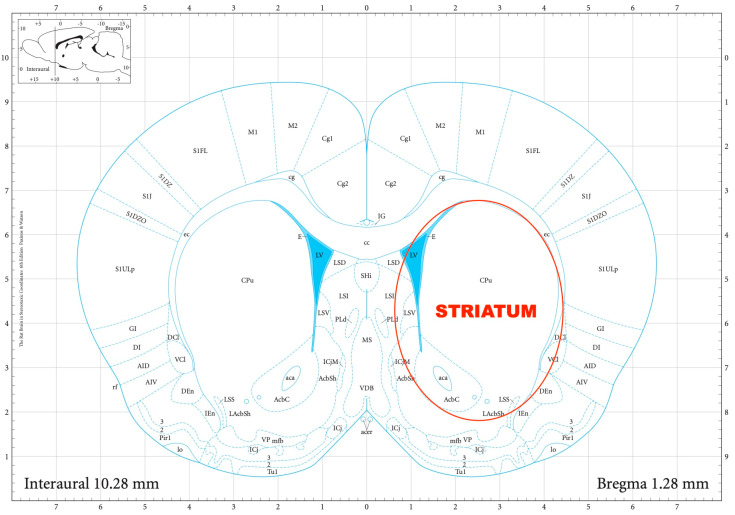
The dissection of the striatum according to a stereotaxic atlas of the rat brain. Reprinted/adapted with permission from Ref. [11]. The coordinates were: rostro-caudal (RC) +4.0–−1.0 mm, medio-lateral (ML) +1.0–+5.0 mm, dorso-ventral (DV) +3.0–+8.0 mm for the striatum.

**Figure 3 biomedicines-11-01288-f003:**
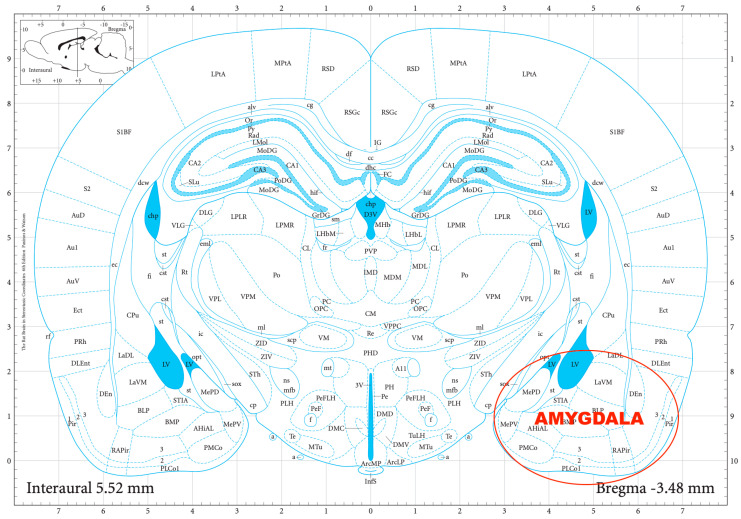
The dissection of the amygdala according to a stereotaxic atlas of the rat brain. Reprinted/adapted with permission from Ref. [11]. The coordinates were: rostro-caudal (RC) 0.0–−2.0 mm, medio-lateral (ML) +3.0–+6.0 mm, dorso-ventral (DV) +7.0–+10.0 mm for the amygdala.

**Figure 4 biomedicines-11-01288-f004:**
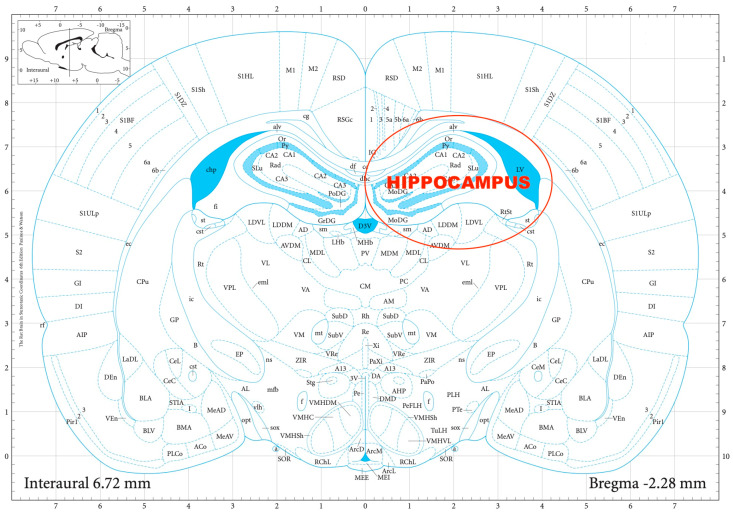
The dissection of the hippocampus according to a stereotaxic atlas of the rat brain. Reprinted/adapted with permission from Ref. [11]. The coordinates were: rostro-caudal (RC) −4.0–−6.0 mm, medio-lateral (ML) +2.0–+5.0 mm, dorso-ventral (DV) +3.0 to +8.0 mm for the hippocampus.

**Figure 5 biomedicines-11-01288-f005:**
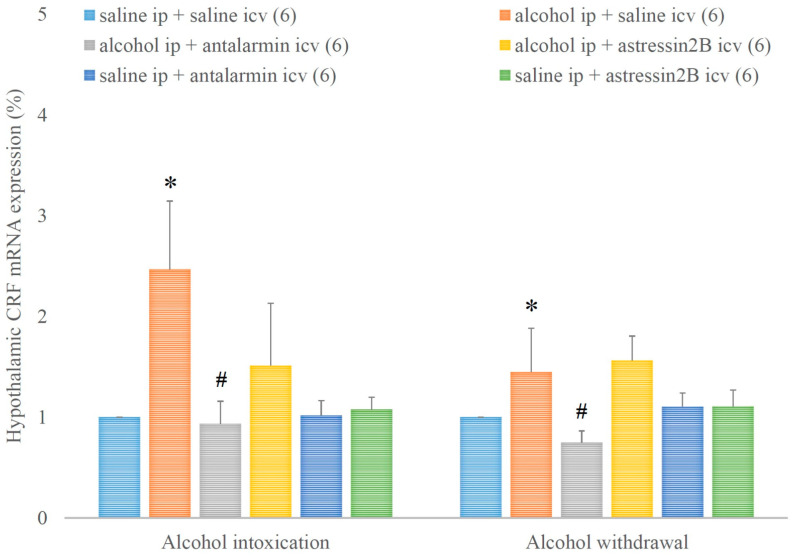
The effects of alcohol intoxication and withdrawal on the hypothalamic corticotropin-releasing factor (CRF) expression in rats and the impacts of antalarmin and astressin_2_B on these effects. Values are presented as the means ± SEM; statistically significant difference was accepted for *p* < 0.05 and indicated with * for alcohol ip + saline icv vs. saline ip + saline icv and with # for alcohol ip + antalarmin icv vs. alcohol ip + saline icv.

**Figure 6 biomedicines-11-01288-f006:**
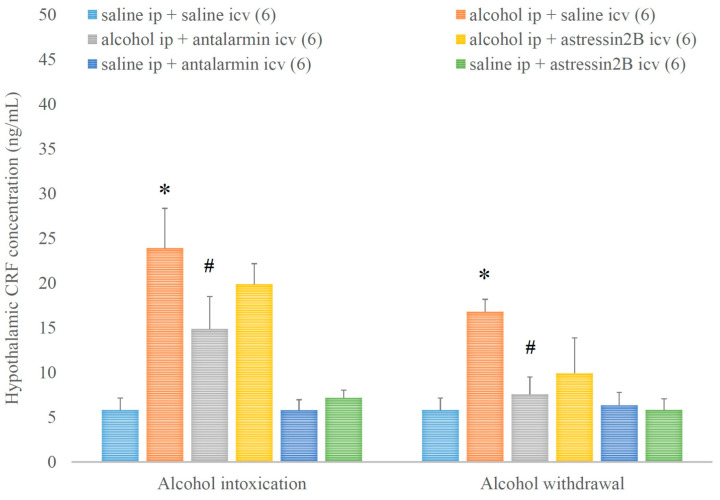
The effects of alcohol intoxication and withdrawal on the hypothalamic corticotropin-releasing factor (CRF) concentration in rats and the impacts of antalarmin and astressin_2_B on these effects. Values are presented as the means ± SEM; statistically significant difference was accepted for *p* < 0.05 and indicated with * for alcohol ip + saline icv vs. saline ip + saline icv and with # for alcohol ip + antalarmin icv vs. alcohol ip + saline icv.

**Figure 7 biomedicines-11-01288-f007:**
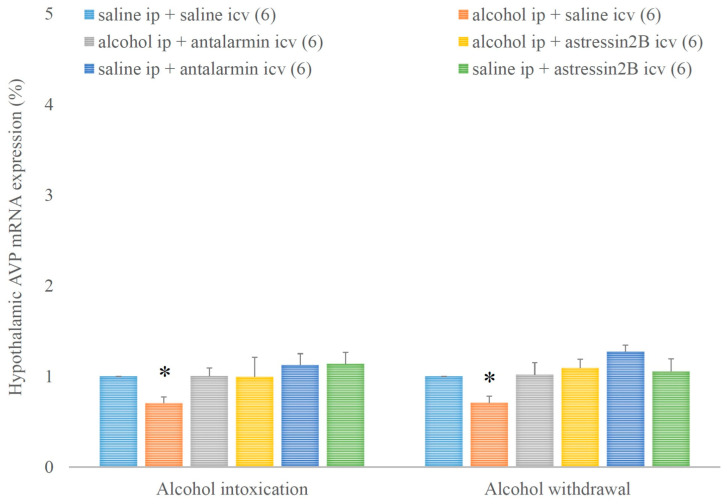
The effects of alcohol intoxication and withdrawal on the hypothalamic arginine vasopressin (AVP) expression in rats and the impacts of antalarmin and astressin_2_B on these effects. Values are presented as the means ± SEM; statistically significant difference was accepted for *p* < 0.05 and indicated with * for alcohol ip + saline icv vs. saline ip + saline icv.

**Figure 8 biomedicines-11-01288-f008:**
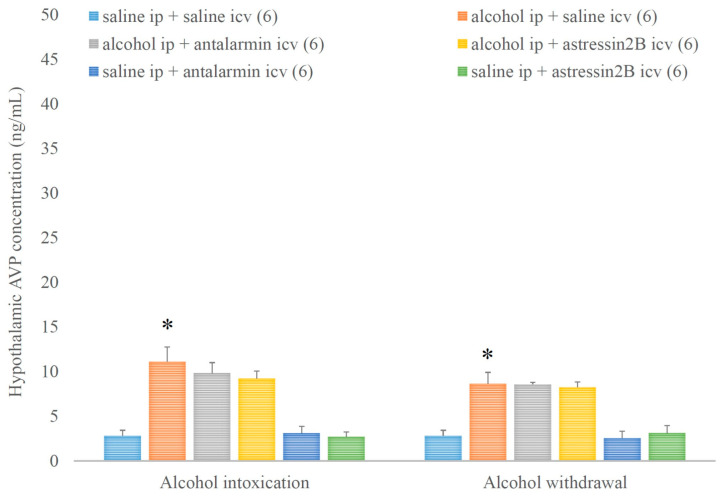
The effects of alcohol intoxication and withdrawal on the hypothalamic arginine vasopressin (AVP) concentration in rats and the impacts of antalarmin and astressin_2_B on these effects. Values are presented as the means ± SEM; statistically significant difference was accepted for *p* < 0.05 and indicated with * for alcohol ip + saline icv vs. saline ip + saline icv.

**Figure 9 biomedicines-11-01288-f009:**
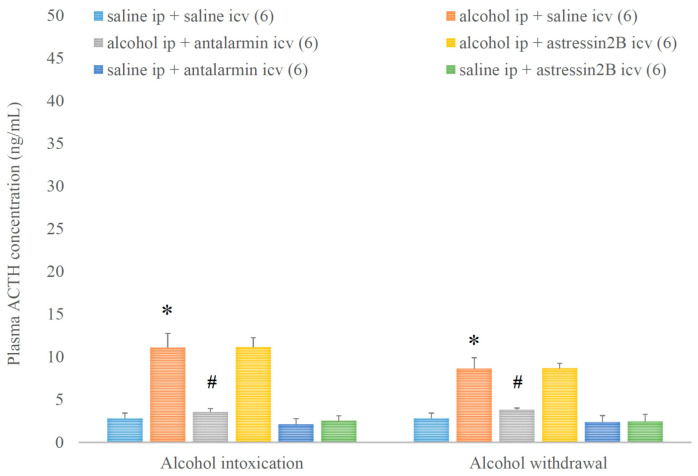
The effects of alcohol intoxication and withdrawal on the plasma adrenocorticoptropic hormone (ACTH) concentration in rats and the impacts of antalarmin and astressin_2_B on these effects. Values are presented as the means ± SEM; statistically significant difference was accepted for *p* < 0.05 and indicated with * for alcohol ip + saline icv vs. saline ip + saline icv and with # for alcohol ip + antalarmin icv vs. alcohol ip + saline icv.

**Figure 10 biomedicines-11-01288-f010:**
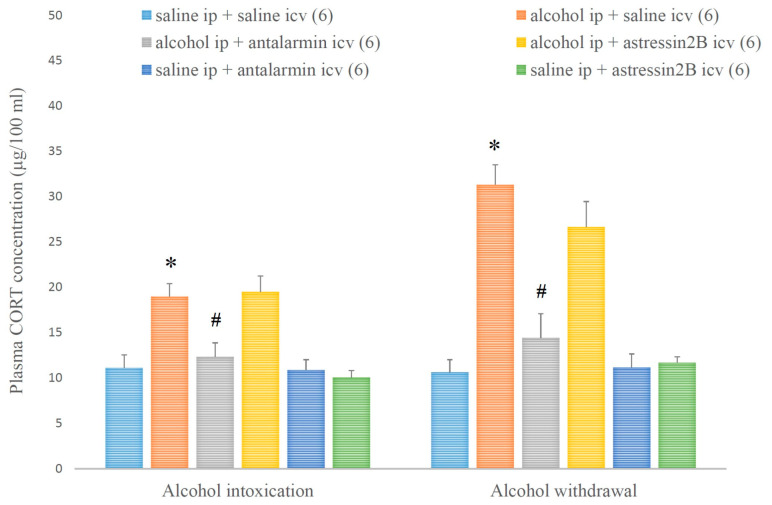
The effects of alcohol intoxication and withdrawal on the plasma corticosterone (CORT) concentration in rats and the impacts of antalarmin and astressin_2_B on these effects. Values are presented as the means ± SEM; statistically significant difference was accepted for *p* < 0.05 and indicated with * for alcohol ip + saline icv vs. saline ip + saline icv and with # for alcohol ip + antalarmin icv vs. alcohol ip + saline icv.

**Figure 11 biomedicines-11-01288-f011:**
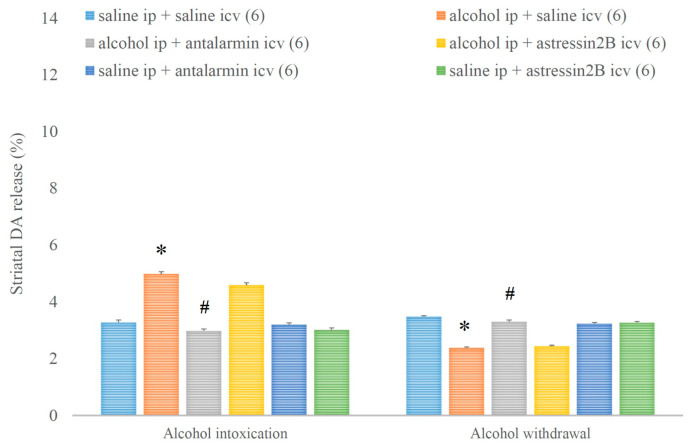
The effects of alcohol intoxication and withdrawal on striatal dopamine (DA) release in rats and the impacts of antalarmin and astressin_2_B on these effects. Values are presented as means ± SEM; statistically significant difference was accepted for *p* < 0.05 and indicated with * for alcohol ip + saline icv vs. saline ip + saline icv and with # for alcohol ip + antalarmin icv vs. alcohol ip + saline icv.

**Figure 12 biomedicines-11-01288-f012:**
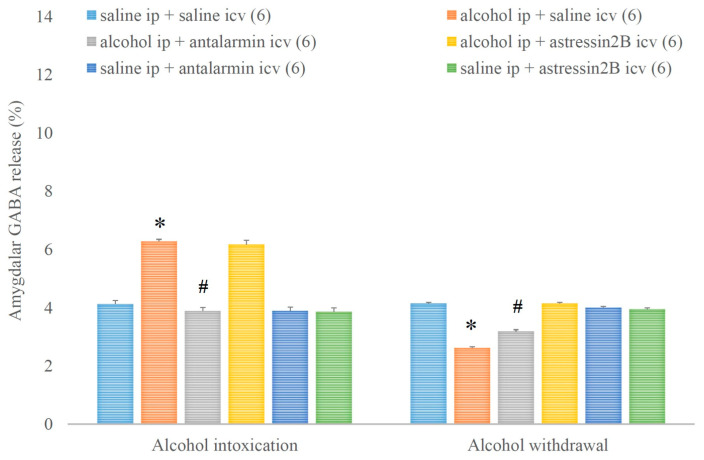
The effects of alcohol intoxication and withdrawal on the amygdalar gamma aminobutyric acid (GABA) release in rats and the impacts of antalarmin and astressin_2_B on these effects. Values are presented as the means ± SEM; statistically significant difference was accepted for *p* < 0.05 and indicated with * for alcohol ip + saline icv vs. saline ip + saline icv and with # for alcohol ip + antalarmin icv vs. alcohol ip + saline icv.

**Figure 13 biomedicines-11-01288-f013:**
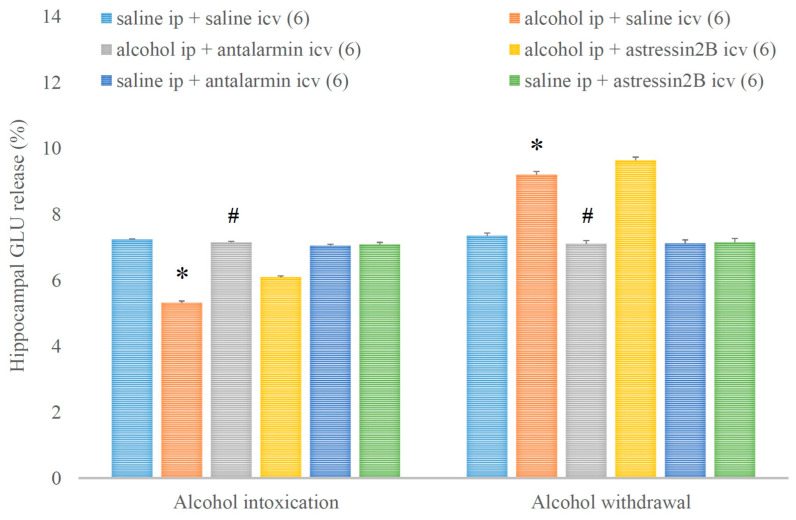
The effects of alcohol intoxication and withdrawal on the hippocampal glutamate (GLU) release in rats and the impacts of antalarmin and astressin_2_B on these effects. Values are presented as the means ± SEM; statistically significant difference was accepted for *p* < 0.05 and indicated with * for alcohol ip + saline icv vs. saline ip + saline icv and with # for alcohol ip + antalarmin icv vs. alcohol ip + saline icv.

**Table 1 biomedicines-11-01288-t001:** The treatment protocol.

Group	Day 1	Day 2	Day 3	Day 4	Day 5	Day 6
Group #1 (*N* = 6)	Saline ip at 8:00 and 20:00	Saline ip at 8:00 and 20:00	Saline ip at 8:00 and 20:00	Saline ip at 8:00 and 20:00	Saline ip + saline icv at 8:00 + assays at 8:30	
Group #2 (*N* = 6)	Saline ip at 8:00 and 20:00	Saline ip at 8:00 and 20:00	Saline ip at 8:00 and 20:00	Saline ip at 8:00 and 20:00	Saline ip + antalarmin icv at 8:00 + assays at 8:30	
Group #3 (*N* = 6)	Saline ip at 8:00 and 20:00	Saline ip at 8:00 and 20:00	Saline ip at 8:00 and 20:00	Saline ip at 8:00 and 20:00	Saline ip + astressin_2_B icv at 8:00 + assays at 8:30	
Group #4 (*N* = 6)	Alcohol ip at 8:00 and 20:00	Alcohol ip at 8:00 and 20:00	Alcohol ip at 8:00 and 20:00	Alcohol ip at 8:00 and 20:00	Alcohol ip + saline icv at 8:00 + assays at 8:30	
Group #4 (*N* = 6)	Alcohol ip at 8:00 and 20:00	Alcohol ip at 8:00 and 20:00	Alcohol ip at 8:00 and 20:00	Alcohol ip at 8:00 and 20:00	Alcohol ip + saline icv at 8:00 + assays at 8:30	
Group #5 (*N* = 6)	Alcohol ip at 8:00 and 20:00	Alcohol ip at 8:00 and 20:00	Alcohol ip at 8:00 and 20:00	Alcohol ip at 8:00 and 20:00	Alcohol ip + antalarmin icv at 8:00 + assays at 8:30	
Group #6 (*N* = 6)	Alcohol ip at 8:00 and 20:00	Alcohol ip at 8:00 and 20:00	Alcohol ip at 8:00 and 20:00	Alcohol ip at 8:00 and 20:00	Alcohol ip + astressin_2_B icv at 8:00 + assays at 8:30	
Group #7 (*N* = 6)	Saline ip at 8:00 and 20:00	Saline ip at 8:00 and 20:00	Saline ip at 8:00 and 20:00	Saline ip at 8:00 and 20:00	Saline ip at 8:00	Saline icv at 8:00 + assays at 8:30
Group #8 (*N* = 6)	Saline ip at 8:00 and 20:00	Saline ip at 8:00 and 20:00	Saline ip at 8:00 and 20:00	Saline ip at 8:00 and 20:00	Saline ip at 8:00	Antalarmin icv at 8:00 + assays at 8:30
Group #9 (*N* = 6)	Saline ip at 8:00 and 20:00	Saline ip at 8:00 and 20:00	Saline ip at 8:00 and 20:00	Saline ip at 8:00 and 20:00	Saline ip at 8:00	Astressin_2_B icv at 8:00 + assays at 8:30
Group #10 (*N* = 6)	Alcohol ip at 8:00 and 20:00	Alcohol ip at 8:00 and 20:00	Alcohol ip at 8:00 and 20:00	Alcohol ip at 8:00 and 20:00	Alcohol ip at 8:00	Saline icv at 8:00 + assays at 8:30
Group #11 (*N* = 6)	Alcohol ip at 8:00 and 20:00	Alcohol ip at 8:00 and 20:00	Alcohol ip at 8:00 and 20:00	Alcohol ip at 8:00 and 20:00	Alcohol ip at 8:00	Antalarmin icv at 8:00 + assays at 8:30
Group #12 (*N* = 6)	Alcohol ip at 8:00 and 20:00	Alcohol ip at 8:00 and 20:00	Alcohol ip at 8:00 and 20:00	Alcohol ip at 8:00 and 20:00	Alcohol ip at 8:00	Astressin_2_B icv at 8:00 + assays at 8:30

**Table 2 biomedicines-11-01288-t002:** The custom primers.

Gene	Forward	Reverse
*CRF*	5′-TGG TGT GGA GAA ACT CAG AGC-3′	5′-CAT GTT AGG GGC GCT CTC TTC-3′
*AVP*	5′-CTG ACA TGG AGC TGA GAC AGT-3′	5′-CGC AGC TCT CGT CGC T-3′
*Gapdh*	5′-CGG CCA AAT CTG AGG CAA GA-3′	5′-TTT TGT GAT GCG TGT GTA GCG-3′

**Table 3 biomedicines-11-01288-t003:** The cycling protocol.

Phase	Temperature °C	Time	Number of Cycles
UDG pre-treatment	50	2 min	1
Initial denaturation	95	10 min	1
Denaturation	95	15 s	40
Annealing	60	30 s	40
Extension	72	30 s	40

## Data Availability

The datasets generated during the current study are available from the corresponding author upon reasonable request.

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
