# Peer review of "The Effects of Alcohol Intoxication and Withdrawal on Hypothalamic Neurohormones and Extrahypothalamic Neurotransmitters"

_biomedicines, 2023, doi:10.3390/biomedicines11051288_

Round 1
Reviewer 1 Report
This is a well-written and scholarly manuscript describing the results of a study examining the role of CRF antagonists on the neural responses to repeated ethanol administration before and after a withdrawal period. The study is well-executed and I only have a few comments on the manuscript.
Treatments: I think a Table outlining the timing of the drug treatments, sacrifice times and assays performed would be helpful. I found the sequence of the procedures hard to follow from the methods. Alternatively, the methods could be presented sequentially, e.g., animals decapitated without anesthesia, trunk blood collected, brains removed, etc.
Design and statistical analyses: I am a little unclear on the statistical approach. It is stated that a 2X2 ANOVA was performed, presumably individually from day 5 and 6 of sacrifice. Even if the design was split across 2 days, shouldn't this be a 2X3 ANOVA (ethanol treatment X drugs/control)? Also, as comparisons are made in the pattern of results across days, it seems that the design is a 2X3X2 ANOVA (ethanol treatment X drugs/control/days). If there is a specific effect of withdrawal, then there should be an interaction with days. Perhaps the problem here is that the study is underpowered to detect such an interaction. It is stated that 72 animals were used in the study. With a 2X3X2 design, that would leave and n=6/group.
Dissections: A figure (based on the rat atlas) showing the location of the dissections would be helpful.
Data availability: It is disturbing that the original data can not be found. Don't scientific rigor and reproducibility require that the original data can be made available? The authors need to explain this.
Author Response
Dear reviewer,
Thank you for your positive response regarding our manuscript entitled: ”The effects of alcohol intoxication and withdrawal on hypothalamic neurohormones and extrahypothalamic neurotransmitters” written by Balázs Simon, András Buzás, Péter Bokor, Krisztina Csabafi, Katalin Eszter Ibos, Éva Bodnár, László Török, Imre Földesi, Andrea Siska, Zsolt Bagosi. We highly appreciate the Reviewers’ comments, based on which we revised our mansucript, including the Figures and the Tables. We would be very grateful if our manuscript in this revised form would be accepted for publication in Biomedicine.
Best wishes,
Zsolt Bagosi
Reviewers’ comments:
Reviewer 1.
This is a well-written and scholarly manuscript describing the results of a study examining the role of CRF antagonists on the neural responses to repeated ethanol administration before and after a withdrawal period. The study is well-executed and I only have a few comments on the manuscript.
1. Treatments: I think a Table outlining the timing of the drug treatments, sacrifice times and assays performed would be helpful. I found the sequence of the procedures hard to follow from the methods. Alternatively, the methods could be presented sequentially, e.g., animals decapitated without anesthesia, trunk blood collected, brains removed, etc.
Response to the Reviewer:
In agreement with the Reviewer, we have added a table (Table 1.) to the revised manuscript, that includes the timing of drug treatments and assays for each group of rats. Consequently, the other tables have been also re-numbered and re-arranged (Table 2-3.).
Also, for a better understanding, the methods have been presented sequentially, and three sentences have been added to the the last paragraph of the Introduction section and Treatment subsection of the Materials and Methods section, as follows:
”After 30 minutes, the mice were decapitated without anesthesia, trunk blood was collected, and the brains were removed. From the brain the expression and concentration of hypothalamic CRF and AVP, and the release of striatal DA, amygdalar GABA and hippocampal GLU were determined. From the trunk blood the concentration of plasma ACTH and CORT were measured.”
2. Design and statistical analyses: I am a little unclear on the statistical approach. It is stated that a 2X2 ANOVA was performed, presumably individually from day 5 and 6 of sacrifice. Even if the design was split across 2 days, shouldn't this be a 2X3 ANOVA (ethanol treatment X drugs/control)? Also, as comparisons are made in the pattern of results across days, it seems that the design is a 2X3X2 ANOVA (ethanol treatment X drugs/control/days). If there is a specific effect of withdrawal, then there should be an interaction with days. Perhaps the problem here is that the study is underpowered to detect such an interaction. It is stated that 72 animals were used in the study. With a 2X3X2 design, that would leave and n=6/group.
Response to the Reviewer: Yes, indeed: the statistical approach was not explained clearly. For this purpose we changed the description of the Statistical analysis in the Methods section as follows:
”Data were presented as means ± SEM. Statistical analysis of the results was performed by ANOVA if test prerequisites allowed using SPSS Software (SPSS Inc., USA). A two-way 2 (alcohol or saline) x 3 (Antalarmin or Astressin2B or saline) ANOVA was performed with estimated marginal means calculated followed by Bonferroni post-hoc test. A probability level of less then 0.05 was accepted as indicating a statistically significant difference”.
Originally, we did perform a three-way ANOVA to include the analysis of the time factor. However, results were discouraging, and the reason behind that might be the small sample size.
3. Dissections: A figure (based on the rat atlas) showing the location of the dissections would be helpful.
Response to the Reviewer: In agreement with the Reviewer, we have added four figures (Figures 1-4.) to the revised manuscript, that includes the dissection of the hypothalamus, striatum, amygdala and hippocampus according to a stereotaxic atlas of the rat brain. Consequently, the other figures have been also re-numbered and re-colored (Figures 5-13.).
4. Data availability: It is disturbing that the original data can not be found. Don't scientific rigor and reproducibility require that the original data can be made available? The authors need to explain this.
Response to the Reviewer: We are sorry for the inconvenience, but we thought not to share the original data publicly. We believe that scientific rigor and reproducibility requires a detailed description of methodology, rather than original data. Nevertheless, we are ready to send the Reviewer the statistical data.
Reviewer 2 Report
Review of manuscript entilted: “The effects of alcohol intoxication and withdrawal on hypothalamic neurohormones and extrahypothalamic neurotransmitters” authored by Balázs Simon, András Buzás, Péter Bokor, Krisztina Csabafi, Katalin Eszter Ibos, Éva Bodnár, László Török, Imre Földesi, Andrea Siska, Zsolt Bagosi
First of all I want to thank you for opportunity to review this interesting manuscript.
In this manuscript, authors investigated changes in several neurotransmitters caused by ethanol intoxication or withdrawal. Introduction provides sufficient information about the undertaken problem. Methods are described extensively, however some details about qPCR, statistical analysis are missing. Results are clearly presented. Discussion is written logically and based on obtained results.
Overall, manuscript is very good, easy to follow and interesting. Some improvements must be done before publishing.
Major concerns:
- Abstract – “The present study provides a new pre-clinical evidence that selective CRF1 antagonists are promising drugs in the treatment of alcohol addiction” – while I understand why authors formulated such a statement but I do not think it is justified to be such confident across the experiments performed in this study. Molecular analysis may show that administration of CRF1 antagonist reverses the effect of ethanol to “baseline” level but we are lacking behavioral observation (e.g. ethanol intake in free choice two-bottle paradigm or assessment of withdrawal symptoms)
- Statistical analysis – were the normality of the data assessed? By which test?
- Methods
- “(…) gene relative to GAPDH was determined (…)”, according to gene nomenclature guidelines should be Gapdh
- primer sequences for Gapdh are missing
Minor concerns:
- Introduction – “Alcohol can be considered a stressor itself, as it (…)”, while I understand that physiological effects of ethanol intake may be similar to those caused by stress, that statement is a little bit controversial in my opinion, since alcohol is often used as stress-relief drug and has anxiolytic and anti-depressive properties as authors mention further in the introduction.
- Figures
- According to y-axis title, numbers are presented as [%]. I believe that it is clearly typo and the scale should be expressed as 100, 200, 300 etc.
- From my personal point of view, I find grey-scale as hard to follow due to presence of six experimental groups, maybe some pattern will be easier to distinguish
Author Response
Dear reviewer,
Thank you for your positive response regarding our manuscript entitled: ”The effects of alcohol intoxication and withdrawal on hypothalamic neurohormones and extrahypothalamic neurotransmitters” written by Balázs Simon, András Buzás, Péter Bokor, Krisztina Csabafi, Katalin Eszter Ibos, Éva Bodnár, László Török, Imre Földesi, Andrea Siska, Zsolt Bagosi. We highly appreciate the Reviewers’ comments, based on which we revised our mansucript, including the Figures and the Tables. We would be very grateful if our manuscript in this revised form would be accepted for publication in Biomedicine.
Best wishes,
Zsolt Bagosi
Reviewers’ comments:
Reviewer 2.
First of all I want to thank you for opportunity to review this interesting manuscript. In this manuscript, authors investigated changes in several neurotransmitters caused by ethanol intoxication or withdrawal. Introduction provides sufficient information about the undertaken problem. Methods are described extensively, however some details about qPCR, statistical analysis are missing. Results are clearly presented. Discussion is written logically and based on obtained results. Overall, manuscript is very good, easy to follow and interesting. Some improvements must be done before publishing.
Major concerns:
1. Abstract – “The present study provides a new pre-clinical evidence that selective CRF1 antagonists are promising drugs in the treatment of alcohol addiction” – while I understand why authors formulated such a statement but I do not think it is justified to be such confident across the experiments performed in this study. Molecular analysis may show that administration of CRF1 antagonist reverses the effect of ethanol to “baseline” level but we are lacking behavioral observation (e.g. ethanol intake in free choice two-bottle paradigm or assessment of withdrawal symptoms).
Response to the Reviewer: We share the Rewiewer’s opinion, therefore the sentence “The present study provides a new pre-clinical evidence that selective CRF1 antagonists are promising drugs in the treatment of alcohol addiction” has been removed from the Abstract and the Conclusions section of the revised manuscript.
2. Statistical analysis – were the normality of the data assessed? By which test?
Response to the Reviewer: Normality was assessed via histogram, kurtosis and skewness (not greater then +/-1 was accepted as normal distribution).
3. Methods – “(…) gene relative to GAPDH was determined (…)”, according to gene nomenclature guidelines should be Gapdh, Primer sequences for Gapdh are missing.
Response to the Reviewer: The gene GAPDH has been changed into Gapdh, according to the gene nomenclature guidelines, and the primer sequences for Gapdh that were missing in the original version, have been added to the table including the custom primers (Table 2.), in the revised version of our manuscript.
Minor concerns:
4. Introduction – “Alcohol can be considered a stressor itself, as it (…)”, while I understand that physiological effects of ethanol intake may be similar to those caused by stress, that statement is a little bit controversial in my opinion, since alcohol is often used as stress-relief drug and has anxiolytic and anti-depressive properties as authors mention further in the introduction.
Response to the Reviewer: We share the Rewiewer’s opinion, therefore the sentence “Alcohol can be considered a stressor itself”, has been removed from the Introduction section of the revised manuscript.
5. Figures: According to y-axis title, numbers are presented as [%]. I believe that it is clearly typo and the scale should be expressed as 100, 200, 300 etc.
From my personal point of view, I find grey-scale as hard to follow due to presence of six experimental groups, maybe some pattern will be easier to distinguish.
Response to the Reviewer: As regards the release of the extrahypothalamic neurotransmitters (e.g., striatal dopamine release), the fractional release was calculated as the ratio between the radioactivity of the samples and that of the remaining brain tissue and this was represented as percentages (%), as mentioned in the Materials and Methods section.
Instead, we have re-colored the Graphic Abstract and the Figures. Also, we have added four figures (Figures 1-4.) to the revised manuscript, that includes the dissection of the hypothalamus, striatum, amygdala and hippocampus according to a stereotaxic atlas of the rat brain. Consequently, the figures have been re-numbered (Figures 1-13.).